# Accelerating Eigenvalue Dataset Generation
# via Chebyshev Subspace Filter

## Abstract

Eigenvalue problems are among the most important topics in many scientific disciplines. With the recent surge and development of machine learning, neural eigenvalue methods have attracted significant attention as a forward pass of inference requires only a tiny fraction of the computation time compared to traditional solvers. However, a key limitation is the requirement for large amounts of labeled data in training, including operators and their corresponding eigenvalues. To tackle this limitation, we propose a novel method, named **S**orting **C**hebyshev **S**ubspace **F**ilter (**SCSF**), which significantly accelerates eigenvalue data generation by leveraging similarities between operators—a factor overlooked by all existing methods. Specifically, SCSF employs truncated fast Fourier transform (FFT) sorting to group operators with similar eigenvalue distributions and constructs a Chebyshev subspace filter that leverages eigenpairs from previously solved problems to assist in solving subsequent ones, reducing redundant computations. To the best of our knowledge, SCSF is the first method to accelerate eigenvalue data generation. Experimental results show that SCSF achieves up to a $6\times$ speedup compared to various numerical solvers.

## 1. Introduction

Solving eigenvalue problems is an important challenge in fields such as quantum physics (Pfau et al., 2023), electromagnetism (Augenstein et al., 2023), and structural mechanics (Wen et al., 2022). Traditional numerical solvers, such as the Krylov-Schur algorithm (Stewart, 2002), often suffer from prohibitively high computational costs when tackling complex problems. To overcome these computational challenges, recent advancements in deep learning (Schütt et al., 2017; Li et al., 2020; Luo et al.) have demonstrated remarkable success as one forward pass only necessitates a tiny fraction of the computation time compared to numerical solvers, often in milliseconds.

Despite their success, data-driven approaches face a fundamental limitation: the reliance on labeled datasets. Training neural networks require large-scale labeled data, which is often generated using computationally expensive traditional methods. For example, the QM9 dataset (Ramakrishnan et al., 2014) contains $1.34 \times 10^5$ molecular data points, each produced by solving Hamiltonian operator eigenvalue problems. These calculations typically employ traditional algorithms such as the Krylov-Schur method (Stewart, 2002), whose computational costs can escalate dramatically with increasing problem complexity, like finer grid resolutions or higher accuracy requirements. This scalability issue represents a significant bottleneck for generating the labeled data needed to train deep learning models. Furthermore, the diversity of scientific problems leads to the need for a unique dataset for each scenario, which further intensifies this challenge of computational intractability. As a result, the high computational expense of generating eigenvalue data severely limits the practical application of deep learning approaches (Zhang et al., 2023).

In particular, the dataset generation process typically involves six key steps, as illustrated in the flowchart in Figure 1. Among these steps, the computation of eigenvalues of matrices is the most computationally demanding (step 4), accounting for 95% of the total processing cost (Hughes, 2012). Existing data generation methods typically compute the eigenvalues of each matrix in the dataset independently. However, the operators in the dataset often share similarities, as they describe related physical phenomena, which can largely simplify and accelerate the eigenvalue-solving process. Existing approaches, however, fail to leverage these similarities, leading to significant computational redundancy. Previous works (Wang et al., 2024; Dong et al., 2024) on solving PDE datasets have demonstrated the potential of leveraging similarity to significantly reduce the time required for dataset generation. However, effectively exploiting matrix similarity to accelerate eigenvalue problem solving for datasets remains a significant challenge.

To address this challenge, we introduce a novel data generation approach, named **S**orting **C**hebyshev **S**ubspace **F**ilter (**SCSF**). SCSF is designed to leverage the approximate eigenpairs of close problems to reduce redundant computa-

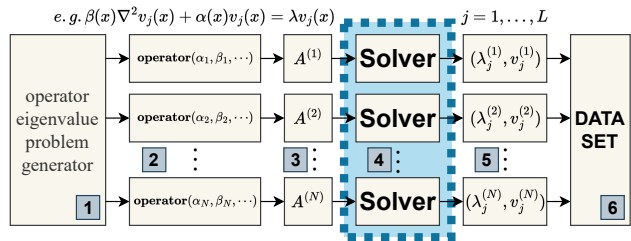

$$e.g.\ \beta(x)\nabla^2 v_j(x) + \alpha(x)v_j(x) = \lambda v_j(x) \qquad j=1,\ldots,L$$

Figure 1: Generation process of the eigenvalue dataset: 1. Generate a set of random problem parameters. 2. Derive the corresponding operators based on these parameters. 3. Convert the operators into matrices using discretization methods. 4. Independently solve for the matrix eigenvalues using numerical solvers. 5. Obtain the matrix eigenpairs, converting them into the operator eigenpairs. 6. Assemble the dataset.

tions in the eigenvalue solving process, thereby accelerating dataset generation. Specifically, in the initial stages, SCSF employs a sorting algorithm based on truncated Fast Fourier transform (FFT), which arranges these operators efficiently, enhancing the adjacent correlation between them and laying the groundwork for sequential solving. Then, SCSF accelerates the convergence of iterations and significantly reduces computation times by constructing a Chebyshev subspace filter, which solves the problem aided by the eigenpairs identified from previous problem solutions. The core design of SCSF is to identify and exploit the close spectral distributions and invariant subspaces within these eigenvalue problems. SCSF coordinates the sequential resolution of these systems rather than treating them as discrete entities. This improved approach not only alleviates the computational demands of eigenvalue solutions but also significantly speeds up the generation of training data for related data-driven algorithms. We summarize our contributions as follow:

- To the best of our knowledge, SCSF is the first method to accelerate the eigenvalue data generation.

- By using truncated FFT sorting and the Chebyshev subspace filtered iteration, we introduce a novel approach that solves the operator problem sequentially.

- Comprehensive experiments demonstrate that SCSF substantially reduces the computational cost of eigenvalue dataset generation. As demonstrated in Figure 2, our method achieves up to a $6\times$ speedup compared to state-of-the-art solvers.

## 2. Related work

### 2.1. Eigenvalue Datasets and Neural Eigenvalue Methods

Eigenvalue datasets are prevalent in neural eigenvalue methods. In quantum chemistry research, eigenvalue algorithms

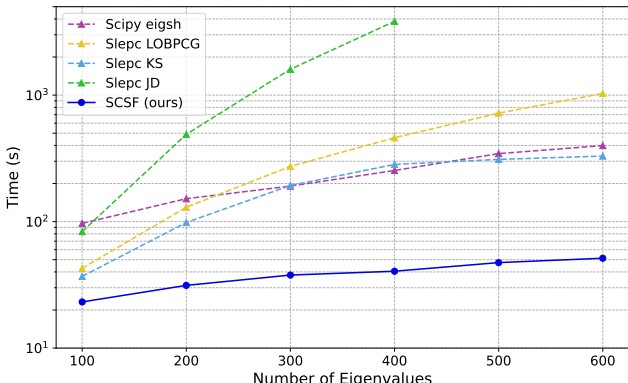

Figure 2: Results of average computation times across various algorithms based on the number of eigenpairs solving.

are commonly used to determine key molecular features, such as orbital energy levels (Kittel & McEuen, 2018). These features, which form the basis of the datasets, are obtained by solving the eigenvalues of Hamiltonian operators (Helgaker et al., 2013). Notable datasets in this domain include QM7 (Blum & Reymond, 2009), QM9 (Ramakrishnan et al., 2014), ANI-1 (Smith et al., 2017), MD17 (Chmiela et al., 2017). These datasets have been widely used to train and validate neural eigenvalue methods (Schütt et al., 2017; Bartók et al., 2017; Rupp et al., 2012), thereby advancing tasks in molecular property prediction and materials design. Besides, Luo et al. accelerates the solution of linear equations by predicting the eigenfunctions of differential operators, which requires a dataset of eigenfunctions for training.

### 2.2. Data Generation for Eigenvalue Algorithms

Training data-driven algorithms require a large amount of labeled eigenvalue data. Typically, the generation of these high-precision data is obtained by traditional algorithms. In the field of computational mathematics, solving operator eigenvalue problems often involves utilizing various discretization methods such as finite difference methods (FDM) (Strikwerda, 2004), finite element methods (FEM) (Hughes, 2012; Johnson, 2012). These discretization methods transform operator eigenvalue problems into matrix eigenvalue problems, which are then solved using the corresponding matrix algorithms. For larger matrices, the Krylov-Schur algorithm (Stewart, 2002), Jacobi-Davidson (Sleijpen & Van der Vorst, 2000), and locally optimal block preconditioned conjugate gradient (LOBPCG) (Knyazev, 2001) are among the most frequently employed algorithms (Golub & Van Loan, 2013).

Nonetheless, traditional methods were not designed for dataset generation, resulting in high computational costs, which have become a significant barrier to the advancement of data-driven approaches (Zhang et al., 2023; Hao et al., 2022). Recent data augmentation research (Brandstetter

et al., 2022; Liu et al., 2023) has led to the development of methods that preserve symmetries and conservation laws, enhancing model generalization and data efficiency. Wang et al. (2024); Dong et al. (2024) report acceleration in the process of solving linear equations, thereby speeding up the generation of PDE datasets. However, these improvements largely focus on neural networks or the rapid solution of linear equation-based PDEs, without discussing optimizations in the generation of eigenvalue datasets.

## 2.3. Chebyshev Filter Technique

The Chebyshev filter technique originates from polynomial approximation theory, where the core concept involves using Chebyshev polynomials to accelerate the convergence of eigenvalues (Zhou & Saad, 2007). This technique constructs a polynomial filter that selectively amplifies spectral components in a specified interval, thereby speeding up the solution of specific eigenvalues. This technique is particularly effective in dealing with sequence eigenvalue problems (Saad, 2011; Zhou et al., 2006a) and has been applied in various contexts, such as stability analysis in electronic structure (Pieper et al., 2016; Banerjee et al., 2016) and quantum chemical computations (Mohr et al., 2017; Zhou et al., 2014; 2006b). To further adapt this technique to the generation of operator eigenvalue datasets, we have developed a specialized sorting algorithm that transforms dataset generation into a sequential solving problem. Throughout the solving process, eigenpairs obtained from previous solutions are used to construct Chebyshev filters, accelerating subsequent solutions.

# 3. Preliminaries

## 3.1. Discretization of Eigenvalue Problem

Our main focus is on solving matrix eigenvalues, the most time-consuming part of data generation. As depicted in Figure 1, these problems are typically solved by using numerical discretization methods such as FDM (Strikwerda, 2004; LeVeque, 2002). These discretization techniques embed the infinite-dimensional Hilbert space of operators into an appropriate finite-dimensional space, thereby transforming operator problems into matrix problems. We provide a simple example to clarify the discussed processes. A detailed account of the equation assembly process can be found in Appendix A. Specifically, we discuss using FDM to solve the eigenvalue problem of the two-dimensional Poisson operator, transforming it into a matrix eigenvalue problem:

$$k(x,y)\nabla^2 u(x,y) = \lambda u(x,y). \tag{1}$$

We map the problem onto a $2 \times 2$ grid (i.e., $N_x = N_y = 2$ and $\Delta x = \Delta y$), where both the variable $u_{i,j}$ and the coefficients $k_{i,j}$ follow a row-major order. This setup facilitates

the derivation of the matrix eigenvalue equation:

$$\begin{bmatrix} k_{1,1} & 0 & 0 & 0 \\ 0 & k_{1,2} & 0 & 0 \\ 0 & 0 & k_{2,1} & 0 \\ 0 & 0 & 0 & k_{2,2} \end{bmatrix} \begin{bmatrix} -4 & 1 & 1 & 0 \\ 1 & -4 & 0 & 1 \\ 1 & 0 & -4 & 1 \\ 0 & 1 & 1 & -4 \end{bmatrix} \begin{bmatrix} u_{1,1} \\ u_{1,2} \\ u_{2,1} \\ u_{2,2} \end{bmatrix} = \lambda \begin{bmatrix} u_{1,1} \\ u_{1,2} \\ u_{2,1} \\ u_{2,2} \end{bmatrix}. \tag{2}$$

By employing various methods to generate the parameter matrices $P$,

$$P = \begin{bmatrix} k_{11} & k_{12} \\ k_{21} & k_{22} \end{bmatrix}. \tag{3}$$

Such as utilizing Gaussian random fields (GRF) or truncated polynomials, we can derive Poisson operators characterized by distinct parameters.

Typically, training a neural network requires $10^3$ to $10^5$ data (Lu et al., 2019). Such a multitude of eigenvalue systems, derived from the same distribution of operators, naturally exhibit a highly similarity (Soodhalter et al., 2020). It is precisely this similarity that is key to the effective acceleration of SCSF. We can conceptualize this as the task of solving a sequential series of matrix eigenvalue problems:

$$A^{(i)}v_j^{(i)} = \lambda_j^{(i)}v_j^{(i)}, \quad j = 1, \cdots, L; \quad i = 1, 2, \cdots, \tag{4}$$

where the matrix $A^{(i)} \in \mathbb{C}^{n \times n}$, the eigenvector $v_j^{(i)} \in \mathbb{C}^n$ and the eigenvalue $\lambda_j^{(i)} \in \mathbb{C}$ vary depending on the operator. We define the eigenpairs as $(\Lambda^{(i)}, V^{(i)})$, with $\Lambda^{(i)} = \text{diag}(\lambda_1^{(i)}, \ldots, \lambda_L^{(i)})$ and $V^{(i)} = [v_1^{(i)} | \cdots | v_L^{(i)}]$.

## 3.2. The Chebyshev Polynomials and Chebyshev Filter

Chebyshev filtered subspace iteration is closely related to Chebyshev orthogonal polynomials (Mason & Handscomb, 2002; Rivlin, 2020). Chebyshev polynomials are widely used due to their strong approximation capabilities. The Chebyshev polynomials $C_m(t)$ of degree $m$ are defined on the interval $[-1, 1]$ and are expressed as

$$C_m(t) = \cos(m \cos^{-1}(t)), \quad |t| \leq 1. \tag{5}$$

$C_m(t)$ commonly referred to as the Chebyshev polynomial of the first kind, satisfies the following recurrence relation:

$$C_{m+1}(t) = 2tC_m(t) - C_{m-1}(t). \tag{6}$$

For a Hermitian matrix $A \in \mathbb{C}^{n \times n}$ and vectors $Y_0 \in \mathbb{C}^{n \times k}$, we use the three-term recurrence relation that defines Chebyshev polynomials in vector form:

$$C_{m+1}(Y_0) = 2AC_m(Y_0) - C_{m-1}(Y_0), \tag{7}$$

$$C_m(Y_0) \equiv C_m(A)Y_0. \tag{8}$$

The computation of $C_m(Y_0)$ and the Chebyshev filter is described in Algorithm 1. Let $A'$ denote the previously

solved related matrix, with $(\lambda_i', v_i')$ in ascending order, and $\{\lambda_2', \ldots, \lambda_n'\} \in [\alpha, \beta]$. In Algorithm 1, the parameter $\lambda$ is typically approximated by $\lambda_1'$, while $c = \frac{\alpha+\beta}{2}$ and $e = \frac{\beta-\alpha}{2}$ represent the center and half-width of the interval $[\alpha, \beta]$, providing estimates for the spectral distribution of $A$.

---

**Algorithm 1:** Chebyshev Filter (Berljafa et al., 2015)

---

**Input:** Matrix $A \in \mathbb{C}^{n \times n}$, vectors $Y_0 \in \mathbb{C}^{n \times k}$ sorted according to ascending degree specification $m = (m_1, \ldots, m_k) \in \mathbb{N}^k$, and parameters $\lambda, c, e \in \mathbb{R}$.

**Output:** Filtered vectors $Y_m = C_m(Y_0)$, where each vector $Y_{m,j}$ is filtered with a Chebyshev polynomial of degree $m_j$.

1   $A = (A - cI_n)/e$;

2   $\sigma_1 = e/(\lambda - c)$;

3   $Y_1 = \sigma_1 A Y_0$;

4   $s = \underset{j=1,\ldots,k}{\mathrm{argmin}} \; m_j \neq 1$;

5   **for** $i = 1, \ldots, m_k - 1$ **do**

6     $\sigma_{i+1} = 1/(2/\sigma_1 - \sigma_i)$;

7     $Y_{i+1,1:s-1} = Y_{i,1:s-1}$;

8     $Y_{i+1,s:k} = 2\sigma_{i+1} A Y_{i,s:k} - \sigma_{i+1}\sigma_i Y_{i,s:k}$;

9     $s = \underset{j=1,\ldots,k}{\mathrm{argmin}} \; m_j \neq i+1$;

---

# 4. Method

In this section, we introduce our novel method, named Sorting Chebyshev Subspace Filter (SCSF), a fast data generation approach that improves the efficiency of solving eigenvalue problems by leveraging intrinsic spectral correlations among operators. SCSF incorporates two key components: (1) a truncated Fast Fourier Transform (FFT)-based approach for efficiently sorting operator eigenvalue samples and (2) the Chebyshev filtered subspace iteration (ChFSI) employed for sequential solving. By integrating these components, SCSF enables effective utilization of prior spectral information, accelerating the eigenvalue data generation.

We first introduce the sorting algorithm that leverages the spectral similarities and provides the time complexity analysis in Section 4.1 . Then we give an introduction to the Chebyshev filtered subspace iteration in Section 4.2. Figure 3 shows the overview of our SCSF. Generally, the truncated FFT sorting algorithm ensures that successive matrices in the sequence exhibit close relations. Then ordered sequence enables the Chebyshev filtered subspace iteration (ChFSI) to effectively utilize prior information, thereby accelerating the solution process.

### 4.1. The Sorting Algorithm

To benefit the successive solving sequence of the eigenvalue problem, we need the sorting algorithm that pulls matrices with similar spectral properties, like frequency series, close enough in the solving sequence, so that eigenvalue solving of the current matrix in sequence can be easily boosted by the last solving. A naive sorting strategy is a greedy sort that uses the similarity of spectral properties between two matrices, like frequency as the distance. And by repeatedly fetching without reservation from the remaining matrix in the data pool, we can re-organize the solving sequence so that the successive solving can benefit from the re-ordered sequence.

---

**Algorithm 2:** The Truncated FFT Sorting Algorithm

---

**Input:** Sequence of eigenvalue problems to be solved $A^{(i)} \in \mathbb{C}^{n \times n}$, corresponding parameter matrix $P^{(i)} \in \mathbb{C}^{p \times p}$, $i = 1, 2, \cdots, N$ and $k$ is the truncation threshold for low frequencies.

**Output:** Sequence for eigenvalue problems $seq_{mat}$.

1   Initialize the list with sequence $seq_0 = \{1, 2, \cdots, N\}$, $seq_{mat}$ is an empty list;

2   Set $i_0 = 1$ as the starting point. And remove 1 from $seq_0$ and append 1 to $seq_{mat}$;

3   **for** $i = 1, \cdots, N$ **do**

4     Let $P_{low}^{(i)} = \mathrm{Trunc}_k\left(\mathrm{FFT}(P^{(i)})\right)$. Perform truncated FFT on matrix $P^{(i)}$ to extract low-frequency information, and $P_{low}^{(i)} \in \mathbb{C}^{k \times k}$;

5   **for** $i = 1, \cdots, N-1$ **do**

6     Refresh $dis$ and set it to a large number, e.g., 1000;

7     **for** *each* $j$ *in* $seq_0$ **do**

8       $dis_j = $ the Frobenius norm of the difference between $P_{low}^{(i_0)}$ and $P_{low}^{(j)}$;

9       **if** $dis_j < dis$ **then**

10        $dis = dis_j$ and $j_{min} = j$;

11     Remove $j_{min}$ from $seq_0$ and append $j_{min}$ to $seq_{mat}$ and set $i_0 = j_{min}$;

12   Get the sequence for eigenvalue problems $seq_{mat}$;

---

However, the core computational cost of such a naive sorting algorithm arises from repeatedly calculating the distances between different matrices $A$, which is directly related to the matrix dimension—that is, the resolution of operators. Recalling Section 3.1, our eigenvalue problem, the matrix $A$, is generated from the parameter matrix $P$ (Lu et al., 2022; Li et al., 2020). Existing works (Holmes, 2012; Li et al., 2020) have shown that the key variables that affect operators stem from the low-frequency components of the parameter matrices $P$, while high-frequency components often represent noise or irrelevant data. Based on this insight, to reduce computational overhead during sorting, we first

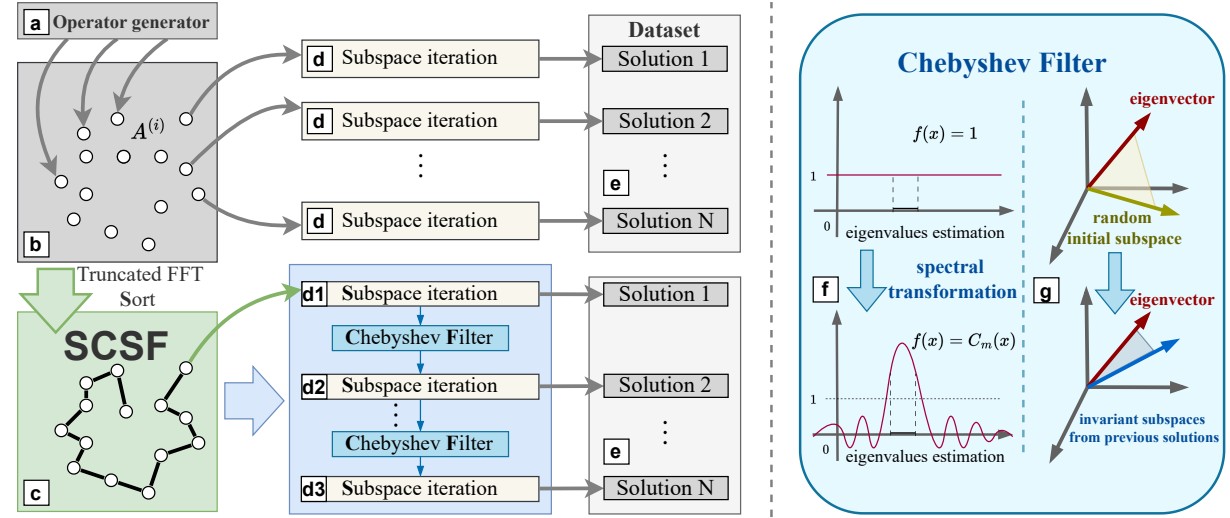

Figure 3: Algorithm Flow Diagram: **a**. Generation of operators to be solved. **b**. Discretization of operators into matrixes. **c**. Apply SCSF algorithm to sort matrixes, obtaining a sequence with strong correlations. **d**. Other algorithms independently solve the eigenvalue problems from step b. **d1, d2, d3**. SCSF algorithm utilizes Chebyshev subspace iterations to sequentially solve the eigenvalue problems. **e**. Assembly of eigenvalue pairs into a dataset. **f**. Amplification of the interval of interest through spectral transformation. **g**. Replacement of initial subspaces with previously solved invariant subspaces.

perform truncated FFT on the parameter matrices to extract the low-frequency information before sorting. We then sort by comparing the distances between these low-frequency components.

As shown in Algorithm 2, suppose we have $N$ eigenvalue problems, the parameter matrices $P^{(i)} \in \mathbb{C}^{p \times p}$, and the low-frequency truncated matrices $P_{low}^{(i)} \in \mathbb{C}^{k \times k}$. The computational complexity of directly using a greedy algorithm is $\mathcal{O}(N^2 p^2)$. Our sorting algorithm's complexity consists of two main parts: 1. FFT Computation: The complexity of FFT is $\mathcal{O}(p^2 \log p)$ per matrix. For $N$ matrices, this totals $\mathcal{O}(N p^2 \log p)$. 2. Greedy Sorting: The subsequent greedy sorting algorithm has a complexity of $\mathcal{O}(N^2 k^2)$.

Overall, the total complexity is $\mathcal{O}(N^2 k^2 + N p^2 \log p)$. Since $k \ll p$ and $p \ll N$, our sorting algorithm effectively reduces computational cost compared to the naive greedy algorithm.

### 4.2. Chebyshev Filtered Subspace Iteration

In a series of eigenvalue problems, inherent correlations often exist between successive systems. We hypothesize that leveraging the eigenpairs $(\Lambda^{(i-1)}, V^{(i-1)})$ of the previous problem $A^{(i-1)}$ can accelerate the iterative convergence of the subsequent system $A^{(i)}$, thereby significantly enhancing computational performance. For various types of operators, the resulting eigenvalue problems produce matrices with distinct structural characteristics. These unique matrix structures align well with the ChFSI method (Manteuffel, 1977; Saad, 2011; Winkelmann et al., 2019; Berljafa et al., 2015).

To verify the effectiveness of our algorithm, we focus on the most common scenario in eigenvalue problems where the operator is self-adjoint; in this case, the corresponding matrix $A$ is Hermitian.

Algorithm 3 outlines the process of ChFSI for solving the $i$-th eigenvalue problem $A^{(i)}$ ($i > 1$) where $L$ eigenvalues are required. The initial approximate invariant subspace $V^{(i-1)}$ and spectral distribution $\Lambda^{(i-1)}$ are derived from the eigenvectors and eigenvalues of the previous problem $A^{(i-1)}$ in the sequence. The parameter $m$ denotes the polynomial degree in the filter function, typically chosen between 10 and 15. For the first eigenvalue problem $A^{(1)}$ in the sequence, the initial iterative subspace $\tilde{V}_0$ and initial spectrum $\tilde{\Lambda}_0$ are randomly generated.

Specifically, ChFSI begins by estimating the upper bound of the eigenvalue spectrum (line 3) using a few Lanczos iterations and known approximate eigenvalues (Zhou & Li, 2011; Saad, 2011). This estimate aids in constructing the subsequent filter function. In line 5, the Chebyshev filter is applied using the vector form of Chebyshev polynomials; details can be found in the Preliminaries section 3.2. After the Chebyshev filtering step, the vector block $\tilde{V}_0$ spanning the invariant subspace may become linearly dependent. To prevent this, orthonormalization is performed (line 6) using QR decomposition based on Householder reflectors. Line 7 computes the Rayleigh quotient of matrix $A^{(i)}$ using the orthonormalized $\tilde{V}_0$, projecting the eigenvalue problem onto a subspace that approximates the desired eigenspace. In line 8, 9, the reduced eigenvalue problem is diagonalized, and the computed eigenvectors

are projected back to the original problem. At the end of the Rayleigh-Ritz step, residuals of the computed eigenvectors are calculated; converged eigenpairs are locked, and non-converged vectors are set to be filtered again (line 10). For each non-converged vector, the optimal degree of the polynomial filter is updated (line 11) based on its residual and approximate eigenvalue.

---

**Algorithm 3:** Chebyshev Filtered Subspace Iteration

**Input:** Eigenvalue problem $A^{(i)}$, eigenpairs $(\Lambda^{(i-1)}, V^{(i-1)})$ of the previous eigenvalue problem $A^{(i-1)}$ where $\Lambda^{(i-1)} = \text{diag}(\lambda_1^{(i-1)} \ldots \lambda_L^{(i-1)})$. Initial filter degree $m_0$.

**Output:** Wanted eigenpairs $(\Lambda^{(i)}, V^{(i)})$.

1  Initialize empty arrays/matrices $(\tilde{\Lambda}, \tilde{V})$, set $\tilde{\Lambda}_0 = \Lambda^{(i-1)}$, and $\tilde{V}_0 = V^{(i-1)}$;

2  Set the filter degrees $(m_1, \ldots, m_L) = (m_0, \ldots, m_0) =: m$;

3  Estimate the largest eigenvalue via Lanczos iteration;

4  **repeat**

5      Apply Chebyshev filter: $\tilde{V}_0 = C_m(\tilde{V}_0)$;

6      Perform QR orthonormalization on $[\tilde{V}|\tilde{V}_0]$;

7      Compute Rayleigh quotient $G = \tilde{V}_0^\dagger A^{(i)} \tilde{V}_0$;

8      Solve the reduced problem $GW = W\tilde{\Lambda}_0$;

9      Update $\tilde{V}_0 = \tilde{V}_0 W$;

10     Lock converged eigenpairs into $(\tilde{\Lambda}, \tilde{V})$;

11     Update filter degrees $(m_1, \ldots, m_k) = m$;

12 **until** *the number of converged eigenpairs $\geq L$*;

13 Return eigenpairs $(\Lambda^{(i)}, V^{(i)}) = (\tilde{\Lambda}, \tilde{V})$ ;

---

Assuming $m$ is the degree of the polynomial, $n$ is the dimension of the matrix $A$, and $f$ is the number of vectors being filtered, the computational complexity per iteration comprises: 1. Chebyshev Filter: $\mathcal{O}(mnf)$ 2. QR Factorization: $\mathcal{O}(nf^2)$ 3. Rayleigh-Ritz Procedure: $\mathcal{O}(nf^2 + f^3)$ 4. Residuals Check: $\mathcal{O}(nf)$ . Since $m \gg 1$, the Chebyshev filtering step is the most computationally intensive.

The acceleration effectiveness of the Chebyshev filtered subspace iteration heavily depends on selecting approximate invariant subspaces and eigenvalues that promote rapid convergence in subsequent iterations. Proper sorting amplifies their impact, reducing the number of iterations required. This underscores the critical importance of the sorting algorithm in our method.

## 5. Experiment

### 5.1. Experimental Settings

To comprehensively assess the performance of our model, denoted as SCSF, against other algorithms, we conducted

extensive experiments, each simulating the generation of an operator eigenvalue dataset. We primarily compared the average computation times across different numbers of eigenvalues solved and various matrix sizes. These tests encompassed three distinct datasets and four mainstream eigenvalue solving algorithms, with SCSF consistently delivering commendable results. The detailed data is available in the Appendix C.1.

**Baseline**. As previously mentioned, our focus revolves around the eigenvalue problem of matrices derived from self-adjoint differential operators, typically consisting of large sparse Hermitian matrices. We benchmarked against the following mainstream algorithms implemented in professional libraries: 1. Eigsh from SciPy (implicitly restarted Lanczos method) (Virtanen et al., 2020; Lehoucq et al., 1998), 2. Locally optimal block preconditioned conjugate gradient (LOBPCG) algorithm from SLEPc (Knyazev, 2001; Hernandez et al., 2009), 3. Krylov-Schur (KS) algorithm from SLEPc (Stewart, 2002), 4. Jacobi-Davidson (JD) algorithm from SLEPc (Sleijpen & Van der Vorst, 2000). For detailed information, please refer to Appendix B.1.

**Datasets**. To explore the adaptability of the algorithm across different matrix types, we delved into three distinct operator eigenvalue problem challenges: 1. Generalized Poisson operator; 2. Second-order elliptic partial differential operator; 3. Helmholtz operator. For a thorough description of the datasets and their generation, please refer to Appendix B.2.

All experiments focus on computing the smallest $L$ eigenvalues in absolute value and their corresponding eigenvectors, which are indicative of the operator eigenvalues and eigenfunctions. For the runtime environment and experimental parameters, refer to Appendix B.3 and B.4. The hyperparameter analysis experiments and the time distribution of each part of SCSF can be found in Appendix C.4 and C.3.

### 5.2. Main Experiment

Table 1 showcases selected experimental data. From this table, we can infer several conclusions: Firstly, across all tests, our SCSF algorithm consistently maintained the lowest computation times. The most significant improvements appeared in the Helmholtz dataset, where SCSF demonstrated speedups of $8\times$, $20\times$, $6\times$, and $95\times$ compared to Eigsh, LOBPCG, KS, and JD algorithms, respectively. These results confirm that SCSF effectively reduces inherent redundancies in sequential eigenvalue problems, substantially accelerating dataset generation.

Secondly, as the number of eigenvalues $L$ solved per matrix increases, the speed advantage of SCSF over other algorithms becomes more pronounced. For instance, on the second-order elliptic operator dataset, when solving for 200 eigenvalues, SCSF is 2.5 times faster than the Krylov-Schur

Table 1: Comparison of average computation times (in seconds) for eigenvalue problems using various algorithms. The first row lists different algorithms, the first column details the datasets including matrix dimensions and solution precisions, and the second column shows the number of eigenvalues $L$ computed for each matrix. The best algorithm is in bold. The symbol '-' denotes data not recorded due to excessive computation times.

| Dataset | $L$ | Eigsh | LOBPCG | KS | JD | SCSF |
|---|---|---|---|---|---|---|
| Poisson | 200 | 14.20 | 73.03 | 23.76 | 270.2 | **12.85** |
| 2500 | 300 | 26.27 | 151.5 | 45.95 | 920.8 | **25.61** |
| 1e-12 | 400 | 36.86 | 265.3 | 72.32 | 2691 | **33.91** |
| Ellipse | 200 | 41.82 | 139.2 | 61.77 | 414.3 | **24.08** |
| 4900 | 300 | 62.47 | 264.1 | 110.5 | 1446 | **29.88** |
| 1e-10 | 400 | 87.19 | 459.7 | 188.7 | 3386 | **34.60** |
| Helmholtz | 200 | 151.7 | 129.9 | 98.34 | 489.6 | **31.31** |
| 6400 | 400 | 253.5 | 460.4 | 283.0 | 3829 | **40.52** |
| 1e-8 | 600 | 398.8 | 1031 | 329.6 | - | **51.32** |

method and 5.5 times faster at 400 eigenvalues. This efficiency stems from SCSF inheriting approximate invariant subspaces from previous solutions, effectively leveraging available information to expand the initial search space. Consequently, SCSF requires minimal additional iterations as $L$ increases, resulting in modest computation time growth.

Thirdly, the performance disparity across different datasets is significant. For example, on the generalized Poisson operator dataset, SCSF is only about 10% faster than Eigsh, yet it leads by 4-7 times on the Helmholtz dataset. This difference can be attributed to the numerical properties of different operators and the matrix assembly formats, which directly influence algorithmic performance.

Furthermore, the impact of the matrix dimension is significant. We conducted supplementary experiments, with the data available in the Appendix C.2. The results are shown in Figure 4, SCSF performs noticeably better as matrix dimensions increase. Below the matrix dimension of 3600, SCSF and Eigsh show comparable efficiency. However, beyond 5000, SCSF significantly outperforms Eigsh and other algorithms. This phenomenon is analyzed from the matrix approximation of operators. For a fixed operator, its eigenvalues and eigenfunctions are fixed. Different matrix dimensions represent embedding the operator in different finite-dimensional linear spaces. For a fixed number of eigenvalues $L$, larger matrices more accurately approximate the true eigenvalues (the smallest $L$ eigenvalues by absolute value) of the operator. In other words, larger matrix dimensions result in fewer errors and noise in the computed eigenvalues, allowing for a clearer demonstration of the similarities between operators. Consequently, larger matrix dimensions allow SCSF to better exploit the similarities,

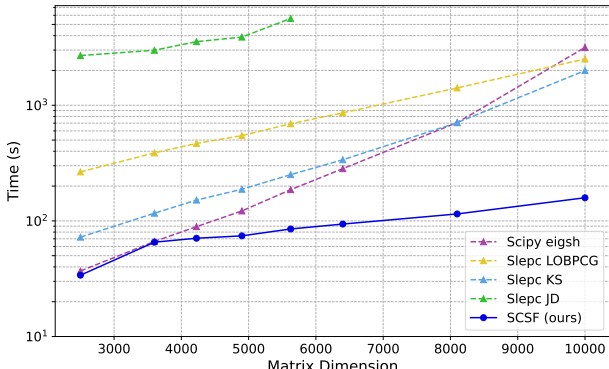

Figure 4: Plot of average computation time versus matrix dimension for solving 400 eigenvalues with a precision of $1e-12$ on the generalized Poisson operator dataset.

yielding superior performance.

### 5.3. Efficacy of Chebyshev Subspace Filter

To analyze the efficacy of the Chebyshev Subspace Filter, we conducted the following experiments. After sorting, the initial vector or subspace for the existing algorithms was set to the eigenvectors from the previous problem. We compared the computational time across different methods. The experiments were conducted on the Helmholtz operator dataset, with a matrix dimension of 6400 and a solution accuracy of $1e-8$. The results are shown in Table 2.

First, the computation time for SCSF in all experiments was minimal, clearly demonstrating the efficacy of the Chebyshev subspace filter. This also highlights that the Chebyshev subspace filter is the optimal choice for leveraging problem similarity to reduce redundancy.

Second, modifying the initial setup had varying impacts on different algorithms. 1. LOBPCG: showed significant acceleration. Its underlying logic is similar to SCSF, both relying on iterative optimization of the subspace to solve the problem. The initial subspace has a considerable impact on the solution. 2. Eigsh and KS were almost unaffected. These methods start with an initial vector and solve the problem through Krylov iteration. In other words, problem similarity only impacts a vector, with little effect on the overall time. 3. JD showed a performance decline. This is because its performance is sensitive to the size of the initial subspace. Our modification altered the default dimension of the initial subspace.

### 5.4. Efficacy of Sorting Algorithms

We analyze the performance of the sorting algorithm module from two perspectives: 1. Comparing the performance of SCSF algorithm with and without the use of 'sorting' as shown in Table 3. 2. Evaluating the effectiveness of different

Table 2: Impact of initial setup modifications on average computation time (in seconds) for different algorithms. '*' denotes the modified initial subspace version. The first-row lists algorithms, and the first column shows the number of eigenvalues $L$ computed. The best algorithm is in bold, and '-' indicates unrecorded data due to excessive computation time.

| $L$ | Eigsh | Eigsh* | LOBPCG | LOBPCG* | KS | KS* | JD | JD* | SCSF (ours) |
|---|---|---|---|---|---|---|---|---|---|
| 200 | 151.7 | 150.2 | 129.9 | 95.9 | 98.34 | 100.6 | 489.6 | 760.1 | **31.31** |
| 300 | 208.8 | 206.3 | 270.1 | 199.8 | 179.9 | 185.2 | 1803 | 3101 | **38.67** |
| 400 | 253.5 | 249.1 | 460.4 | 362.1 | 283.0 | 292.2 | 3829 | 6374 | **40.52** |
| 500 | 324.6 | 315.3 | 717.3 | 573.7 | 314.2 | 317.4 | - | - | **46.70** |
| 600 | 398.8 | 394.7 | 1031 | 866.0 | 329.6 | 335.7 | - | - | **51.32** |

Table 3: Performance comparison of SCSF with and without sorting. The first column lists the number of eigenvalues $L$ computed, while subsequent columns display average computation times, average iteration counts, total Flop counts, and filter Flop counts. Experiments used the matrix dimension of 2500 and precision $1e-12$ on the generalized Poisson operator dataset.

| $L$ | Time (s) | | Iteration | | Flops | | Filter Flops | |
|---|---|---|---|---|---|---|---|---|
| | nosort | sort | nosort | sort | nosort | sort | nosort | sort |
| 20 | 8.248 | 2.971 | 19.70 | 9.880 | 519.7 | 298.4 | 485.8 | 280.8 |
| 100 | 14.18 | 9.891 | 18.77 | 35.38 | 1984 | 2332 | 1798 | 1970 |
| 200 | 18.45 | 12.85 | 36.30 | 33.67 | 4459 | 3944 | 3654 | 3192 |
| 300 | 34.59 | 25.61 | 47.50 | 39.18 | 8967 | 7544 | 6985 | 5702 |
| 400 | 42.60 | 33.91 | 47.43 | 45.18 | 12022 | 11182 | 9087 | 8338 |

Table 4: Comparison of average computation times (in seconds) for different sorting algorithms, with the first column indicating dataset size. Experiments used the matrix dimension of 6400 on the Helmholtz dataset.

| Size | Greedy | Truncated FFT Sort (ours) | | |
|---|---|---|---|---|
| | Total | FFT | Greedy | Total |
| $10^2$ | 0.114 | 0.0016 | 0.0147 | 0.0163 |
| $10^3$ | 7.328 | 0.0164 | 1.421 | 1.438 |
| $10^4$ | 592.7 | 0.1658 | 150.9 | 151.1 |

Table 5: Comparison of average computation times (in seconds) and iteration counts for different sorting algorithms using SCSF. Experiments used the matrix dimension of 6400 on the Helmholtz dataset, precision $1e-8$, and targeting 400 eigenvalues.

| | Nosort | Greedy | Ours |
|---|---|---|---|
| Time (s) | 66.66 | 40.52 | 40.52 |
| Iteration | 10.4 | 5.5 | 5.5 |

sorting algorithms as detailed in Tables 4 and 5.

Firstly, Table 3 indicates that incorporating sorting can enhance SCSF computation speed to 1.3 to 2.8 times, reduce the number of iterations by 5% to 50%, and decrease total Flops by 7% to 43%. The optimization effect of sorting is more pronounced with smaller numbers of solutions ($L$). This is because when $L$ is large, the inherited subspace already contains most of the necessary correlation information, diminishing the impact of sorting. Moreover, the Flops in the Filter component constitute over 70% of SCSF's computational load. A detailed time analysis of different aspects of SCSF can be found in Appendix C.3.

Secondly, as shown in Table 4, our designed truncated FFT sorting algorithm incurs significantly lower time overhead compared to the complete greedy sorting in SKR (Wang et al., 2024), with its benefits becoming more pronounced

as dataset size increases. In the truncated FFT sorting algorithm, the FFT contributes minimally to computational overhead but significantly reduces the time required for subsequent greedy sorting. Table 5 shows SCSF solution times for matrices sorted using either greedy or truncated FFT sorting are nearly identical, highlighting its effectiveness.

## 6. Conclusions

In this paper, we introduced SCSF algorithm. To the best of our knowledge, this is the first method to accelerate eigenvalue dataset generation by reducing computational redundancy in the associated matrix eigenvalue problems. The proposed SCSF algorithm significantly reduces the computational overhead of eigenvalue dataset generation, thereby addressing a major obstacle to the application of neural networks in scientific computing.

## Impact Statement

This paper presents work whose goal is to advance the field of Machine Learning. There are many potential societal consequences of our work, none of which we feel must be specifically highlighted here.

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

# A. From Differential Operator to Matrix Eigenvalue Problem: An Example

## A.1. Overview

The general methodology for solving the eigenvalue problems of differential operators numerically, employing techniques such as Finite Difference Method (FDM), Finite Element Method (FEM), and Spectral Method, can be delineated through the following pivotal steps (Strikwerda, 2004; Hughes, 2012; Johnson, 2012; LeVeque, 2002):

1. Mesh Generation: This step involves dividing the domain, over which the differential operator is defined, into a discrete grid. The grid could be composed of various shapes, including squares, triangles, or more complex forms depending on the problem's geometry.

2. Operator Discretization: The differential operator is transformed into its discrete counterpart. Essentially, this maps the operator from an infinite-dimensional Hilbert space to a finite-dimensional representation.

3. Matrix Assembly: In this phase, the discretized operator is represented in a matrix form. For linear differential operators, this involves creating a system of matrix eigenvalue problems. For nonlinear operators, iterative methods akin to Newton's iteration are employed, transforming the problem into a sequence of matrix eigenvalue problems.

4. Applying Boundary Conditions: This involves discretizing and applying boundary conditions specific to the differential operator in question, which are then incorporated into the matrix system.

5. Solving the Matrix Eigenvalue Problem: This stage, often the most computationally intensive, entails solving the matrix for its eigenvalues and eigenvectors, which correspond to the eigenvalues and eigenfunctions of the original differential operator.

6. Obtaining the Numerical Solution: The final step involves mapping the obtained numerical solutions back onto the original domain, analyzing them for accuracy and stability, and interpreting them in the context of the initial problem.

## A.2. Example

To illustrate how the FDM can transform the wave equation into a system of matrix eigenvalue problems, let's consider a concrete and straightforward example. Assume we aim to solve a one-dimensional wave equation's operator eigenvalue problem, expressed as

$$-\frac{d^2u}{dx^2} = \lambda u, \tag{9}$$

over the interval $[0, L]$. The boundary conditions are $u(0) = u(L) = 0$, signifying fixed-end conditions. In this context, $u(x)$ denotes the eigenfunction, and $\lambda$ represents the eigenvalue.

1. Mesh Generation: Using the central difference quotient, we divide the interval $[0, L]$ into $N + 1$ evenly spaced points, including the endpoints. The distance between adjacent points is denoted as $\Delta x = \frac{L}{N}$.

2. Operator Discretization: This step involves formulating the difference equation. At each interior node, which excludes the endpoints and totals $N - 1$ points, we apply a central difference approximation for the second derivative, represented as

$$\frac{d^2u}{dx^2} \approx \frac{u(x_{i+1}) - 2u(x_i) + u(x_{i-1})}{(\Delta x)^2} \tag{10}$$

3. Matrix Assembly: In this phase, the discretized operator is represented in a matrix form. Following the approximation, we construct the matrix $A$, an $N - 1 \times N - 1$ tridiagonal matrix, crucial for the computations. The matrix $\boldsymbol{A}$ is constructed as:

$$\boldsymbol{A} = \frac{1}{(\Delta x)^2} \begin{bmatrix} -2 & 1 & 0 & \cdots & 0 \\ 1 & -2 & 1 & \cdots & 0 \\ 0 & 1 & -2 & \cdots & 0 \\ \vdots & \vdots & \vdots & \ddots & \vdots \\ 0 & 0 & 0 & \cdots & -2 \end{bmatrix} \tag{11}$$

4. Applying Boundary Conditions: For the wave equation with boundary conditions $u(0) = u(L) = 0$, these fixed-end conditions are integrated into the matrix equation. In the FDM framework, the values at the endpoints ($u_0$ and $u_N$) are zero,

directly reflecting the boundary conditions. The impact of these conditions is encapsulated in the matrix $\boldsymbol{A}$, affecting the entries related to $u_1$ and $u_{N-1}$ (the grid points adjacent to the boundaries). The tridiagonal matrix $\boldsymbol{A}$ incorporates these boundary conditions, ensuring that the computed eigenfunctions satisfy $u(0) = u(L) = 0$.

5. Solving the Matrix Eigenvalue Problem: The final computational step involves solving the matrix eigenvalue problem, expressed as $\boldsymbol{A}\boldsymbol{u} = \lambda\boldsymbol{u}$. This includes determining the eigenvalues $\lambda$ and corresponding eigenvectors $\boldsymbol{u}$, which are discrete approximations of the eigenfunctions of the original differential equation.

6. Obtaining the Numerical Solution: By solving the eigenvalue problem, we obtain numerical solutions that approximate the behavior of the original differential equation. These solutions reveal the eigenvalues and eigenvectors and provide insights into the physical phenomena modeled by the equation.

# B. Details of Experimental Setup

## B.1. Baseline

The baseline algorithms were implemented using the following numerical computing libraries:

- Eigsh: A SciPy (v1.14.1) implementation wrapping ARPACK's SSEUPD and DSEUPD functions, which compute eigenvalues and eigenvectors using the Implicitly Restarted Lanczos Method. Default parameters were used.

- Locally Optimal Block Preconditioned Conjugate Gradient (LOBPCG): Implemented in SLEPc (v3.21.1) with default parameters.

- Krylov-Schur (KS): Implemented in SLEPc (v3.21.1) with default parameters.

- Jacobi-Davidson (JD): Implemented in SLEPc (v3.21.1). The implementation uses 'bcgsl' as the linear equation solver, 'bjacobi' as the preconditioner, and sets the linear equation solving precision to 1e-5.

## B.2. Dataset

1. Generalized Poisson Operator

We consider two-dimensional generalized Poisson operators, which can be described by the following equation (Li et al., 2020; Rahman et al., 2022; Kovachki et al., 2021; Lu et al., 2022):

$$-\nabla \cdot (K(x,y)\nabla h(x,y)) = \lambda h(x,y),$$

In our experiment, $K(x,y)$ is derived using the Gaussian Random Field (GRF) method. We convert these operators into matrices using the central difference scheme of FDM. The parameters inherent to the GRF serve as the foundation for our sort scheme.

2. Second-Order Elliptic Partial Differential Operator

We consider general two-dimensional second-order elliptic partial differential operators, which are frequently described by the following generic form (Evans, 2022; Bers et al., 1964):

$$\mathcal{L}u \equiv a_{11}u_{xx} + a_{12}u_{xy} + a_{22}u_{yy} + a_1 u_x + a_2 u_y + a_0 u = \lambda u,$$

where $a_0, a_1, a_2, a_{11}, a_{12}, a_{22}$ are constants, and $f$ represents the source term, depending on $x, y$. The variables $u, u_x, u_y$ are the dependent variable and its partial derivatives. The equation is classified as elliptic if $4a_{11}a_{22} > a_{12}^2$.

In our experiments, $a_{11}, a_{22}, a_1, a_2, a_0$ are uniformly sampled within the range $(-1, 1)$, while the coupling term $a_{12}$ is sampled within $(-0.01, 0.01)$. We then select equations that satisfy the elliptic condition to form our dataset. We convert these operators into matrices using the central difference scheme of FDM. The coefficients $a_0, a_1, a_2, a_{11}, a_{12}, a_{22}$ serve as the foundation for our sort scheme.

3. Helmholtz Operator

We consider two-dimensional Helmholtz operators, which can be described by the following equation (Zhang et al., 2022):

$$\nabla \cdot (p(x,y)\nabla u(x,y)) + k^2(x,y) = \lambda u(x,y),$$

Physical Contexts in which the Helmholtz operator appears: 1. Acoustics; 2. Electromagnetism; 3. Quantum Mechanics.

In Helmholtz operators, $k$ is the wavenumber, related to the frequency of the wave and the properties of the medium in which the wave is propagating. In our experiment, $p(x, y)$ and $k(x, y)$ are derived using the GRF method. The parameters inherent to the GRF serve as the foundation for our sort scheme.

**B.3. Environment**

To ensure consistency in our evaluations, all comparative experiments were conducted under uniform computing environments. Specifically, the environments used are detailed as follows:

- Platform: Docker version 4.33.1 (windows 11)

- Operating System: Ubuntu 22.04.3 LTS

- Processor: CPU AMD Ryzen 9 8945HS w, clocked at 4.00 GHz

**B.4. Experimental Parameter Configuration**

All baseline methods were implemented using their default parameters from respective libraries.

For SCSF, the following configurations were adopted:

- The size of the inherited subspace varies according to the number of eigenvalues to be computed. Specifically, when calculating 20, 100, 200, 300, and 400 eigenvalues, the corresponding subspace sizes are set to 4, 20, 40, 60, and 80, respectively.

- The filter degree parameter $m$ is consistently set to 20 across all experiments.

## C. Experimental Data and Supplementary Experiments

### C.1. Main Experimental Data

As shown in Tables 7, 6, 8, SCSF showed the best performance among all tested configurations

Table 6: Comparison of average computation times (in seconds) for eigenvalue problems using various algorithms on generalized Poisson operator dataset. The first row lists different algorithms, and the first column shows the number of eigenvalues $L$ computed for each matrix. Matrix Dimension = 4900, Precision = $1e - 10$.

| $L$ | Eigsh | LOBPCG | KS | JD | SCSF (ours) |
|-----|-------|--------|------|------|-------------|
| 150 | 9.15 | 46.8 | 14.9 | 138 | 7.95 |
| 200 | 14.2 | 73.0 | 23.8 | 270 | 12.9 |
| 250 | 19.8 | 109 | 34.3 | 553 | 19.0 |
| 300 | 26.3 | 152 | 45.6 | 921 | 25.7 |
| 350 | 31.5 | 203 | 58.4 | 1732 | 29.8 |
| 400 | 36.9 | 265 | 72.3 | 2691 | 33.9 |
| 450 | 42.8 | 342 | 87.3 | 3708 | 38.3 |

Table 7: Comparison of average computation times (in seconds) for eigenvalue problems using various algorithms on second-order elliptic operator dataset. The first row lists different algorithms, and the first column shows the number of eigenvalues $L$ computed for each matrix. Matrix Dimension = 2500, Precision = $1e-12$.

| $L$ | Eigsh | LOBPCG | KS | JD | SCSF (ours) |
|---|---|---|---|---|---|
| 150 | 31.35 | 91.80 | 40.65 | 214.80 | 19.62 |
| 200 | 41.82 | 139.20 | 61.77 | 414.30 | 24.08 |
| 250 | 52.17 | 197.04 | 84.65 | 861.44 | 28.00 |
| 300 | 62.47 | 264.10 | 110.50 | 1446.00 | 29.88 |
| 350 | 74.59 | 355.18 | 147.01 | 2324.88 | 31.52 |
| 400 | 87.19 | 459.70 | 188.70 | 3386.00 | 34.60 |
| 450 | 100.28 | 577.67 | 235.56 | 4629.38 | 40.05 |

Table 8: Comparison of average computation times (in seconds) for eigenvalue problems using various algorithms on Helmholtz operator dataset. The first row lists different algorithms, and the first column shows the number of eigenvalues $L$ computed for each matrix. Matrix Dimension = 6400, Precision = $1e-8$. The symbol '-' denotes data not recorded due to excessive computation times.

| $L$ | Eigsh | LOBPCG | KS | JD | SCSF (ours) |
|---|---|---|---|---|---|
| 200 | 151.70 | 129.90 | 98.34 | 489.60 | 31.31 |
| 300 | 190.84 | 273.08 | 192.88 | 1601.08 | 37.78 |
| 400 | 253.50 | 460.40 | 283.00 | 3829.00 | 40.52 |
| 500 | 344.60 | 720.33 | 310.21 | - | 47.41 |
| 600 | 398.80 | 1031.00 | 329.60 | - | 51.32 |

**C.2. Analysis of the Influence of Matrix Dimension**

Table 9: Comparison of different algorithms' computation time (in seconds) for varying matrix dimensions using the generalized Poisson operator dataset. Results show average computation times for solving 400 eigenvalues with a precision of $1e-12$.

| Matrix Dimension | Eigsh | LOBPCG | KS | JD | SCSF (ours) |
|---|---|---|---|---|---|
| 2500 | 36.86 | 265.30 | 72.32 | 2691.00 | 33.91 |
| 3600 | 66.41 | 387.20 | 116.50 | 2990.00 | 65.41 |
| 4225 | 89.13 | 467.74 | 151.36 | 3548.13 | 70.79 |
| 4900 | 121.90 | 546.20 | 187.80 | 3886.00 | 74.23 |
| 5625 | 186.21 | 691.83 | 251.19 | - | 85.11 |
| 6400 | 282.80 | 860.00 | 337.70 | - | 93.86 |
| 8100 | 707.95 | 1412.54 | 707.95 | - | 114.82 |
| 10000 | 3162.28 | 2511.89 | 1995.26 | - | 158.49 |

As demonstrated in Table 9, the impact of matrix dimension on algorithm performance reveals several key insights. For matrices below dimension 3600, SCSF and Eigsh show comparable efficiency. However, SCSF's advantages become increasingly pronounced as matrix dimensions grow larger. At dimension 10000, SCSF achieves remarkable speedups: 20x faster than Eigsh, 16x faster than LOBPCG, and 13x faster than KS. This scaling behavior can be attributed to how larger matrix dimensions result in fewer errors and noise in the computed eigenvalues, allowing SCSF to better exploit similarities between problems. Additionally, the JD algorithm becomes computationally intractable at and above dimension 5625, while

SCSF maintains stable performance scaling even at high dimensions.

## C.3. Analysis of Computational Times for SCSF Components

Table 10: Analysis of Computational Times (in seconds) for SCSF Components

| All | Lanczos (line3) | Filter (line 5) | QR (line 6) | RR (line 7) | Resid (line 8, 9) | Sort |
|---|---|---|---|---|---|---|
| 9.89e+0 | 4.04e-2 | 7.41e+0 | 3.12e-1 | 9.76e-1 | 7.95e-1 | 1.51e-2 |

We conducted a statistical analysis of the average time consumption for each component of the SCSF algorithm on the generalized Poisson operator dataset, with a matrix dimension of 2500 and the number of eigenvalues to be solved set to 100. The results are presented in Table 10. The notation "line x" within parentheses corresponds to line x in Algorithm 3, "ALL" denotes the total time consumption, and "sort" represents the average time required by the sorting algorithm. It is evident that the filtering process accounts for over 70% of the total time consumption, which aligns with our theoretical analysis in Section 4.2.

## C.4. Analysis of Hyperparameters

Table 11: Average Computational Times (in seconds) of SCSF under Different Degree Parameters $m$.

| Deg | 12 | 16 | 20 | 24 | 28 | 32 | 36 | 40 |
|---|---|---|---|---|---|---|---|---|
| Time (s) | 43.92 | 39.79 | 40.52 | 40.64 | 40.85 | 41.13 | 41.19 | 43.50 |

We investigated the impact of different degree parameters $m$ on the performance of SCSF. As shown in Table 11, the experiments were conducted on the Helmholtz operator dataset with a matrix dimension of 6400, a solution accuracy of 1e-8, 400 eigenvalues to be solved, and an inherited subspace size of 80. The degree parameter $m$, as described in Algorithm 3, primarily controls the order of the Chebyshev polynomial. The results indicate that varying $m$ within the range of 12 to 40 has a minimal effect on the computation time of SCSF. Therefore, as long as $m$ is chosen within a reasonable range, its specific value does not significantly influence the performance. In the main experiments of this paper, $m$ is fixed at 20.

Table 12: Average Computational Times (in seconds) of SCSF under Different Subspace Dimension.

| Dim | 50 | 60 | 70 | 80 | 90 | 100 | 110 | 120 |
|---|---|---|---|---|---|---|---|---|
| Time (s) | 43.28 | 44.35 | 42.43 | 40.52 | 39.65 | 37.43 | 38.28 | 38.58 |

We examine the influence of different inherited subspace sizes on the performance of SCSF. As presented in Table 12, the experiments are conducted on the Helmholtz operator dataset with a matrix dimension of 6400, a solution accuracy of 1e-8, 400 eigenvalues to be computed, and a degree parameter $m$ set to 20.

The results demonstrate that as the inherited subspace size increases, the computation time of SCSF initially decreases and then rises, reaching its minimum around a size of 100. The reduction in computation time at the front end is attributed to the enriched initial subspace with more available information as the inherited subspace grows. Conversely, the increase in computation time at the back end is due to the significantly higher overhead of performing Chebyshev filtering with a larger inherited subspace.

Overall, as long as the inherited subspace size is set within a reasonable range, its impact on SCSF remains minimal. In our experiments, we consistently set the inherited subspace size to 20% of the number of eigenvalues to be computed.

