# OpenReview forum: "Accelerating Eigenvalue Dataset Generation via Chebyshev Subspace Filter"
_ICML.cc/2025/Conference — Submitted to ICML 2025_

### Official Review · Reviewer_oCdy · 2025-03-11

**Overall Recommendation:** 3

**Summary:**

The paper introduces the Sorting Chebyshev Subspace Filter (SCSF), which is the first method designed to accelerate eigenvalue dataset generation. Instead of solving each eigenproblem independently, SCSF exploits spectral similarities in two key ways. First, it applies a truncated FFT sorting algorithm to reorder the dataset so that matrices with similar eigenvalue distributions are processed sequentially. Second, it uses a Chebyshev filtered subspace iteration that reuses eigenvectors from a previously solved matrix as the initial subspace for the next one. Although recent works on PDE datasets have hinted at speedups by exploiting similarity, SCSF is the first to apply such ideas to eigenvalue computations with a novel algorithmic twist. The approach builds on known concepts—FFT for feature extraction and Chebyshev polynomial filters for eigen-solvers—but integrates them uniquely to achieve large performance gains.

Pros:
1.The paper introduces SCSF, the first method specifically designed to accelerate eigenvalue dataset generation by exploiting spectral similarities.
2. It combines known techniques (FFT and Chebyshev filters) in a unique way to create a sequential, similarity-exploiting process that leads to significant speedups.

Cons:
1. The method’s core contribution is in numerical linear algebra and scientific computing, not in developing new machine learning algorithms
2. The paper lacks an end-to-end demonstration showing how improved data generation benefits subsequent machine learning tasks.
3. The approach assumes that matrices in the dataset have sufficient spectral similarity, but it does not address cases where this assumption might fail.

**Claims And Evidence:**

Most claims in the submission are supported by clear and convincing evidence, particularly through extensive experimental comparisons.

However, some claims are less convincingly supported:

Generality of Spectral Similarity: The paper assumes that matrices in the dataset exhibit sufficient spectral similarity to benefit from reusing eigenvectors. While this assumption is plausible in many practical scenarios, the submission does not fully explore or provide evidence for cases where the spectral similarity is weak. This leaves some uncertainty about the method’s general applicability.

Impact on End-to-End ML Applications: Although the method is motivated by the need to accelerate data generation for machine learning, the submission does not present an end-to-end demonstration of how the faster dataset generation directly improves ML model training or performance. Thus, the claim of significant impact on ML applications is indirectly supported rather than being directly evidenced.

**Essential References Not Discussed:**

NA

**Experimental Designs Or Analyses:**

One minor issue is that the experiments focus exclusively on the eigenvalue computation aspect, without an end-to-end demonstration showing the impact on subsequent machine learning tasks. Additionally, the reliance on spectral similarity as a premise is assumed rather than exhaustively tested across datasets with varying degrees of similarity.

**Methods And Evaluation Criteria:**

The proposed methods and evaluation criteria are well-aligned with the problem at hand.

**Other Comments Or Suggestions:**

Some references are still in preprint format; for example, DeepONet should have been published in Nature Machine Intelligence.

**Other Strengths And Weaknesses:**

NA

**Questions For Authors:**

1. Could you clarify how SCSF performs when the assumption of high spectral similarity among matrices is not met? For example, if the dataset contains matrices with more diverse eigenvalue distributions, how does this affect the convergence speed and overall efficiency?

2. Can you provide an end-to-end demonstration or case study showing how the accelerated eigenvalue dataset generation impacts downstream machine learning tasks (e.g., training neural eigenvalue models)?

3. Could you provide further theoretical analysis or empirical insights into how the truncated FFT sorting algorithm quantitatively improves the reordering of matrices and accelerates the convergence of the Chebyshev filtered subspace iteration?

**Relation To Broader Scientific Literature:**

Overall, the paper integrates well-established ideas—FFT for feature extraction and Chebyshev filtering for eigen-solvers—with the novel concept of sequentially exploiting spectral similarity.

**Theoretical Claims:**

The paper does not include formal proofs for new theoretical claims. Instead, it builds on well-established techniques—such as FFT-based feature extraction and Chebyshev filtered subspace iteration.

---

> ### Author Rebuttal · Authors · 2025-04-01
>
> We sincerely thank the reviewer for their insightful and constructive feedback. Below we respond to your comments as follows and sincerely hope that our rebuttal could properly address your concerns. If so, we would deeply appreciate it if you could raise your score and your confidence. If not, please let us know your further concerns, and we will continue actively responding to your comments and improving our work.
> ## Summary
> > The method’s core contribution is in numerical linear algebra and scientific computing...
>
> We proposes SCSF, an algorithm designed to accelerate eigenvalue dataset generation. **Reducing the computational cost of dataset generation is critically important for AI advancements, particularly in scientific machine learning**. Below, we highlight key evidence supporting the significance of this task:
>
> - Motivation:
>   - As discussed in our Introduction (Section 2), efficient dataset generation is a bottleneck for AI-driven scientific computing.
>   - The AI4Science review ([1], Section 9.2.9 and Section 7.5.2) notes: *"A primary limitation of learned solvers is the requirement of sufficient* *training data* *generated by expensive numerical solvers."*
>   - **Leading ML conferences have published related work emphasizing the importance of efficient dataset generation for AI applications** (e.g., ICML 2022 [2], ICLR Spotlight 2024 [3], ICML 2024 [4]).
> - Key Insight:  Traditional numerical linear algebra focuses on optimizing single-problem solvers, whereas dataset generation requires solving multiple related problems—**a challenge unique to AI-driven scientific computing**.  Our work leverages problem correlations to accelerate dataset generation, designing SCSF specifically for this task.
> ## Q1
> - We clarify that **SCSF does not require global spectral similarity—only that locally nearby problems (in parameter space) share spectral structure**.
>
> - This is justified because:  **Scientific eigenvalue problems derive from physical equations, where eigenvalues/vectors vary continuously with parameters.**  If parameters and spectra were uncorrelated, ML models would fail to learn the mapping—rendering dataset generation moot.
>
> - Empirical Analysis:  When spectral similarity is weak, SCSF degrades to a Chebyshev subspace iteration (equivalent to disabling filtering). We demonstrate this on the Poisson dataset;  precision 1e − 12; $L = 100$; Matrix dimension 2500:
>
>   - |          | SCSF | SCSF (no sort) | SCSF (rand fliter) | Chebyshev subspace iteration |
>     | -------- | ---- | -------------- | ------------------ | ---------------------------- |
>     | time (s) | 9.9  | 14.2           | 14.8               | 14.8                         |
>
>   - ‘rand filter’ randomly generates an invariant subspace for filtering each time the solution is solved, simulating the situation where these problems have no spectral similarity at all.
>
>   - we can see that when there is no spectral similarity, SCSF will degenerate to Chebyshev subspace iteration. This situation is similar to SCSF without sort. This indirectly shows the importance of sorting algorithm for SCSF.
>
> ## Q2
> - We agree that evaluating dataset quality is crucial. Below, we provide theoretical and empirical evidence that SCSF-generated data does not compromise ML training:
>
> 1. **Theoretical Guarantees**: SCSF is only an acceleration algorithm, not a data augmentation algorithm. Under specified error tolerances, SCSF does not alter the generated dataset’s validity. In numerical linear algebra, "ground truth" eigenvalues are typically replaced by high-precision numerical solutions (e.g., with errors < 1e−8). Our experiments meet this threshold, which far exceeds standard ML data requirements.
>
> 2. **Experimental Validation**:  We trained a Fourier Neural Operator to predict the smallest eigenfunction of the Helmholtz operator using two datasets (10,000 samples, rel. error < 1e−8): 1.SCSF 2. SciPy eigs.  Results show identical convergence for both datasets, with training dynamics and final errors (relative L2 error) matching exactly:
>
>    1. |        | 0     | 200   | 400   | final |
>       | ------ | ----- | ----- | ----- | ----- |
>       | eigs 1 | 0.93 | 0.031 | 0.010 | 0.008 |
>       | eigs 2 | 0.92 | 0.032 | 0.011 | 0.008 |
>       | SCSF 1 | 0.82 | 0.038 | 0.012 | 0.008 |
>       | SCSF 2 | 0.92 | 0.028 | 0.010 | 0.008 |
>
>    2. To mitigate randomness, we repeated training twice per dataset. The results confirm that **SCSF does not affect MLmodel performance**.
>
> - Conclusion: SCSF accelerates data generation **without sacrificing downstream ML utility**.
> ## Q3
> Please see the response to reviewer **dk1n's Q2**
>
> [1] Artificial intelligence for science in quantum, atomistic, and continuum systems
>
> [2] Lie Point Symmetry Data Augmentation for Neural PDE Solvers
>
> [3] Accelerating data generation for neural operators via krylov subspace recycling
>
> [4] Accelerating PDE Data Generation via Differential Operator Action in Solution Space

---

> > ### Comment · Reviewer_oCdy · 2025-04-02
> >
> > Thank you for your detailed responses. However, as I pointed out in the first review round, my primary concern remains the connection between this work and the ML community. While the proposed method is an effective acceleration technique for data generation, it is still unclear whether the generated data can truly benefit ML models.
> >
> > The authors have cited various works discussing scientific ML models, but no direct evidence has been provided to demonstrate that their method improves ML model performance. Therefore, I find the response unsatisfactory in addressing this key concern.
> >
> > A potential improvement would be to apply the proposed algorithm to generate training data for a specific ML model and evaluate whether it leads to performance gains on the given task. This would better substantiate the method's practical impact on ML applications.

---

> > > ### Author Response · Authors · 2025-04-02
> > >
> > > Thank you once again for your feedback. Please allow me to address your concerns in detail.
> > >
> > > 1. **Our algorithm accelerates eigenvalue dataset generation without affecting data quality or ML training outcomes** (as clarified in Rebuttal **Q2**). The focus of this work is solely on speeding up dataset generation. Below, I will elaborate on its implications for ML.
> > >
> > > 2. **The key contribution to ML lies in significantly reducing the computational cost of dataset generation for the same problem.** For example, in our Helmholtz dataset case (10,000 samples, matrix dimension 6,400, precision 1e-8, 600 eigenvalues), conventional methods (e.g., SciPy’s `eigsh`) require ~46 days, whereas our SCSF method completes it in just 5 days. Unlike traditional AI domains (e.g., NLP, CV), AI4Science often requires regenerating datasets for different physical scenarios, severely hindering AI deployment in scientific research (eigenvalue problems are central to many scientific applications).
> > >
> > > 3. **Consider an ML practitioner training a model for eigenvalue problems:** If no precomputed dataset exists, generating data might take 46 days (vs. maybe 1 week for model training). With SCSF, this drops to 5 days—**without compromising data quality or model performance**. This time saving allows researchers to train the model five additional times (such as hyperparameter tuning). This efficiency directly facilitates ML research in this domain.
> > >
> > > 4. **For large-scale benchmarks:** Generating 20 distinct physics problems (such as each taking 46 days) would require 800 days. SCSF reduces this to 100 days. Isn’t this transformative?
> > >
> > > 5. **Precedent in AI conferences:** As noted in our Rebuttal **summary**, many top-tier AI works focus on accelerating *linear system* dataset generation for AI4Science (ICLR 2024 spotlight [1], ICML 2024 [2]). Ours is the **first to target eigenvalue datasets**, which are equally critical.
> > >
> > > 6. SCSF is specifically optimized for eigenvalue dataset generation tasks. It leverages intrinsic correlations among problems within the dataset to accelerate computation. Notably, **the requirement for generating large-scale homogeneous datasets is unique to machine learning applications** - such demand simply doesn't exist in traditional computational mathematics domains.
> > >
> > > [1] Accelerating data generation for neural operators via krylov subspace recycling
> > >
> > > [2] Accelerating PDE Data Generation via Differential Operator Action in Solution Space
> > >
> > > We welcome further discussion. If this response resolves your concerns, we sincerely hope you might consider raising the score. Thank you!

---

### Official Review · Reviewer_dk1n · 2025-03-13

**Overall Recommendation:** 1

**Summary:**

The authors propose a method for accelerating the generation of a training dataset for subsequent solution of eigenvalue problems by machine learning methods. The proposed method is based on the idea of using spectral correlations between operators corresponding to different samples (FFT-based approach for sorting and Chebyshev filtered subspace iteration for solving).

**Claims And Evidence:**

Yes

**Essential References Not Discussed:**

References to relevant works are provided and discussed.

**Experimental Designs Or Analyses:**

Yes

**Methods And Evaluation Criteria:**

Yes

**Other Comments Or Suggestions:**

--

**Other Strengths And Weaknesses:**

The work is neatly structured and written in a very consistent and clear manner. However, I have a few questions regarding scientific novelty, which are outlined below.

**Questions For Authors:**

1. I would ask you to formulate more clearly what exactly you consider to be the main innovation proposed in your approach. After studying the manuscript, I got the impression that you are proposing to supplement the known method "ChASE: Chebyshev Accelerated Subspace iteration Eigensolver for sequences of Hermitian eigenvalue problems" (2018) by sorting the generated matrices corresponding to the samples of the data set. In this case, it seems that the scientific novelty of the work is insufficient.

2. You point out the advantage of computational complexity of FFT Sorting Algorithm compared to the greedy approach. But how does this complexity directly relate to the theoretical complexity of solving the main problem via Chebyshev filtered subspace iteration?

**Relation To Broader Scientific Literature:**

Please, see questions below.

**Theoretical Claims:**

Yes

---

> ### Author Rebuttal · Authors · 2025-04-01
>
> We sincerely thank the reviewer for their insightful and constructive feedback. Below we respond to your comments as follows and sincerely hope that our rebuttal could properly address your concerns. If so, we would deeply appreciate it if you could raise your score and your confidence. If not, please let us know your further concerns, and we will continue actively responding to your comments and improving our work.
>
> ## **Q1**
>
> - The core innovations of this paper are as follows：
>   - As stated in the abstract, SCSF introduces a two-stage acceleration framework: 1.Sorting: Rearranges the dataset into a sequence of strongly correlated problems.  2.Chebyshev Filtering: Exploits this correlation to accelerate eigenvalue computation. **These components are mutually dependent—neither suffices alone.**
>   -  (see Intro, Line 41 right side) the original eigenvalue data set generation algorithm solves each problem independently. **SCSF is the first work to optimize dataset generation by leveraging inter-problem correlations via Chebyshev filtering.**
> - Relationship to ChASE:
>   - Differences:
>     - Algorithmic Structure: SCSF incorporates a novel **Truncated FFT Sort module** (Section 2.3, Line 134, left side), absent in ChASE.
>     - Problem Scope: ChASE targets sequences of continuously varying eigenvalue systems (e.g., nonlinear eigenvalue problems [1]). ChASE cannot be directly used for the problem of eigenvalue dataset generation (there is a significant performance loss, see below).
>   - Similarities:
>     - Both exploit spectral approximations via Chebyshev filtering (Section 4.2, Line 272, left side). We explicitly cite ChASE [1] and related works.
>   - Summary: SCSF is a specialized variant of Chebyshev-filtered subspace methods, optimized for dataset generation. **Without sorting, SCSF reduces to ChASE-like performance.**
> - Experiment Comparison with ChASE:
>   - Section 5.4 (Line 424, left side) tests SCSF with/without sorting. Without sorting, SCSF matches ChASE’s performance.
>   - Key Results (Table 3, Line 397):
>     - Sorting improves SCSF’s speed by 1.3–2.8×, reduces iterations by 5–50%. **This fully demonstrates the performance gap between SCSF and ChASE in the dataset generation task.**
>     - Because the gap between tasks in the dataset is large, directly using the Chebyshev subspace iteration algorithm does not work well. These problems need to be sorted first and processed into a sequence with correlations before the Chebyshev filter technology can be effectively accelerated. This point is also mentioned in the abstract of [2].
>
> ## **Q2**
>
> - Necessity of Sorting: As shown in Section 5.4 (Line 424) and Q1, Chebyshev subspace iteration cannot directly accelerate the eigenvalue dataset generation task without significant performance degradation. The sorting module is essential to achieve speedup.  Thus, **complexity analysis of SCSF must account for sorting overhead.**
>
> - Computational Overhead of Sorting: Our experiments in Section 5.4 (Line 436) and Table 4 (Line 411) show that sorting a dataset of size $10^4$ using the greedy approach takes 592s—a prohibitive cost. By contrast, SCSF’s Truncated FFT Sort reduces this overhead to 1/4 of the original time.
>
> - To further address your concern, we provide additional analysis of how sorting enhances dataset quality:
>
>   - We measure the similarity between matrices by computing the cosine of the principal angles between their 10-dimensional invariant subspaces (spanned by the smallest 10 eigenvectors in modulus). Smaller values indicate higher similarity (one-sided distance).
>
>   - Setup: Matrix dimension = 6400 (Helmholtz dataset), 10k samples, precision = 1e-8, 400 eigenvalues computed, parameter matrix $P$ with dimension $p=80$, and varying truncation frequencies $k$ (SCSF default $k=20$).
>
>     - |                        | No sort | k = 10 | k = 20 | k = 30 | k = 40 | Greedy |
>       | ---------------------- | ------- | ------ | ------ | ------ | ------ | ------ |
>       | one-sided distance     | 0.95    | 0.89   | 0.85   | 0.85   | 0.85   | 0.85   |
>       | Sort time (s)          | 0       | 110    | 151    | 193    | 246    | 593    |
>       | Average solve time (s) | 66.7    | 52.2   | 40.5   | 40.5   | 40.5   | 40.5   |
>
>   - Key Findings
>
>     - Sorting **significantly increases inter-problem correlation** in the dataset (explaining the performance gain).
>     - The truncation parameter \(k\) affects sorting time, sorting quality, and solver time. For $k \geq 20$, solver time becomes stable, showing diminishing returns. **This reflects the interplay between sorting and Chebyshev iteration.**
>
> - We will include a more detailed theoretical analysis in the final version to further solidify these insights.
>
> [1] ChASE: Chebyshev Accelerated Subspace iteration Eigensolver for sequences of Hermitian eigenvalue problem
>
> [2] Accelerating data generation for neural operators via krylov subspace recycling, ICLR spotlight 2024

---

### Official Review · Reviewer_3gRC · 2025-03-15

**Overall Recommendation:** 3

**Summary:**

In this work, the authors propose new computational method for accelerating the problems of generating datasets that involve eigenvalue computations. Such problems often appear in applications of deep learning to scientific computing, especially with differential operators, where it is necessary to solve eigenvalue problems for discretized matrices many times. Classical algorithms from numerical linear algebra might take substantial time when solving each such problem independently. The new method is called SCSF (Sorting Chebyshev Subspace Filter) and it is based on two key components:

1. Using the Truncated FFT for a set of generated matrices and a greedy sorting algorithm, which arranges problems in the order where matrices with similar spectra are close to each other.

2. Sequentially solving eigenvalue problems with Chebyshev Filtred Subspace Iteration, employing the solution to the previous problem for every next one.

Numerical experiments demonstrate substatial improvement in terms of the total computational time, when using the new method, as compared to previous approaches, as well as effiiciency of each proposed component.


## update after rebuttal

I appreciate the authors' response. After the rebuttal, I keep my score.

**Claims And Evidence:**

--

**Essential References Not Discussed:**

--

**Experimental Designs Or Analyses:**

--

**Methods And Evaluation Criteria:**

--

**Other Comments Or Suggestions:**

--

**Other Strengths And Weaknesses:**

I think the proposed method is very interesting and can be useful for generating large datasets with eigenvalue problems in practical tasks, that involve differential operators. The paper is generally well written and the numerical results seem substantial.


I have only the following list of small remarks.


1. My main concern is some missing details in the presentation, that make it a bit more difficult to understand the method for a broad audience. For example, Section 3.2 about the Chebyshev Filter (correspondingly Algorithm 1) can benefit from a more detailed explanation on why the Chebyshev Filter is useful in this case. Now, the authors only provide some formulas without further discussion. I.e. it would be useful to mention what happens with vector $C_m(Y_0)$ when $m$ grows. Additionally, the authors should specify $C_0$ and $C_1$ to correctly define the recursion in (6).


2. Could you please clarify what is the parameter $L$ used in (4)? Can it be $L = n$ (the dimension of the problem) or $L << n$?


3. I think the descriptions of Algorithm 2 and Algorithm 3 can also benefit significantly from more clarity. Here is the list of the corresponding concerns:

- Line 2 in Algorihm 2: 'And remove 1 .. and append 1.' --- it's probably better to rephrase.

- It would be better to define 'Trunc_k( )' and 'FFT( )' operations used in Line 4 of Algorithm 2 explicitely, and describe their properties in details.

- Line 6 in Algorithm 2: it is not very clear what does it mean to 'refresh' the variable dis.

- Algorithm 3, Line 3: 'Estimate the largest eigenvalue via Lanczos iteration' -- is it the description of the following part or is it a separate operation? In latter case, it is not clear what is an input and output. I think it would be benefitial to make the description of this part more explicit.

- Algorithm 3, Line 12: 'until the number of converged eigenpairs $\geq  L$' -- it is not very clear how to check this condition.

- The following notation used in Algorithm 3 is not very transparent: '$(m_1, ..., m_L) = (m_0, ..., m_0) = m$'
Does it mean that the constant value of m is assigned to all values of $m_0, m_1, ..., m_L$ or something else? I think it is very confusing in its current form and should be reformulated.


4. I would also appreciate if the authors could add some discussion about the following implementation details regarding the algorithms:

- Now Algorithm 2 uses the 'Greedy Sorting'. Is it possible to improve its complexity further by employing a more advanced sorting algorithm?

- Can the peformance of Algorithm 3 be further enhanced if it also receives the truncated FFT of the input matrix computed previously in Algorithm 2?

- Instead of Lanzcos iterations in Algorithm 3, is it possible to use a simple Power method?

**Questions For Authors:**

--

**Relation To Broader Scientific Literature:**

--

**Theoretical Claims:**

There are no theoretical guarantees for the proposed method. However, empirical results demonstrate good performance.

---

> ### Author Rebuttal · Authors · 2025-04-01
>
> We sincerely thank the reviewer for their insightful and constructive feedback. Below we respond to your comments as follows and sincerely hope that our rebuttal could properly address your concerns. If so, we would deeply appreciate it if you could raise your score and your confidence. If not, please let us know your further concerns, and we will continue actively responding to your comments and improving our work.
>
> ## 1
> > For example, Section 3.2 about the Chebyshev Filter (correspondingly Algorithm 1) can benefit from a more detailed explanation on why the Chebyshev Filter is useful in this case. Now, the authors only provide some formulas without further discussion. I.e. it would be useful to mention what happens with vector Cm(Y0) when m grows. Additionally, the authors should specify C0 and C1 to correctly define the recursion in (6).
>
> - Thank you for the suggestion. Chebyshev polynomials are orthogonal function polynomials, and similar effects could theoretically be achieved with other orthogonal polynomials (e.g., Legendre). However, we chose Chebyshev polynomials due to their superior approximation properties and broad applicability.
> - Here, $m$ denotes the order of the Chebyshev polynomial $C_m(t)$. Higher-order polynomials generally yield better approximation accuracy for $C_m(Y_0)$, but computational costs also increase with $m$. Thus, a trade-off is necessary—we empirically set $m = 20$ for balance.
> - We apologize for the lack of clarity and will include a more detailed discussion in the final version.
>
> ## 2
> > Could you please clarify what is the parameter L used in (4)? Can it be L=n (the dimension of the problem) or L<<n?
>
> - $ L $ represents the number of target eigenvalues. For large sparse matrices derived from differential operators, $L$ is generally much smaller than $n$.
> - We regret the oversight and will add a precise description in the final version.
>
> ## 3
> > Here is the list of the corresponding concerns: ...
>
> Thank you for the detailed feedback. We will revise the algorithms accordingly in the final version. Below are clarifications:
>
> - $ Trunc_k() $ truncates the Fourier transform to retain the lowest $ k $ frequencies, while $FFT() $ denotes the Fast Fourier Transform.
> - $dis$ (Line 8, Algorithm 2) measures the Frobenius distance between two $ P $ matrices.
> - Algorithm 3, Line 3: The input is the current matrix $A$; the output is an approximation of the largest eigenvalue via a few Lanczos iterations.
> - Algorithm 3, Line 12: We evaluate the relative residual of eigenpairs. Due to space constraints, this was omitted; a detailed pseudocode will be added to the appendix in the final version.
> - The notation assigns the constant $m_0$ to all $m_i$. We will rephrase this to avoid ambiguity.
>
> ## 4.1
> > Now Algorithm 2 uses the 'Greedy Sorting'. Is it possible to improve its complexity further by employing a more advanced sorting algorithm?
>
> - Yes—this is a direction for future work. We plan to combine advanced sorting techniques with our 'Truncated FFT Sorting' to further optimize efficiency.
>
> ## 4.2
> > Can the peformance of Algorithm 3 be further enhanced if it also receives the truncated FFT of the input matrix computed previously in Algorithm 2?
>
> - No. Our work focuses on parameterized eigenvalue problems. The relationship between these parameters and matrix elements to be solved is very complex. So, in numerical linear algebra, such parameters cannot directly aid eigenvalue computation.
>
> ## 4.3
> > Instead of Lanzcos iterations in Algorithm 3, is it possible to use a simple Power method?
>
> While possible, Lanczos is preferable for two reasons:
>
> 1. Convergence: Lanczos (a high-order Krylov method) converges faster than the first-order Power Method.
> 2. Efficiency: For Hermitian matrices (arising from self-adjoint operators, as noted in Line 245), Lanczos is a symmetric-optimized variant of Arnoldi iteration and incurs lower computational overhead.

---

### Decision · Program_Chairs · 2025-05-01

**Decision:**

Reject

**Comment:**

Overall, while the proposed method is very interesting and the paper is generally well written, the reviewers raised several concerns regarding the scope of its scientific novelty and how the method can directly improve ML model training.